# Potential Drug Candidates to Treat TRPC6 Channel Deficiencies in the Pathophysiology of Alzheimer’s Disease and Brain Ischemia

**DOI:** 10.3390/cells9112351

**Published:** 2020-10-24

**Authors:** Veronika Prikhodko, Daria Chernyuk, Yurii Sysoev, Nikita Zernov, Sergey Okovityi, Elena Popugaeva

**Affiliations:** 1Laboratory of Molecular Neurodegeneration, Peter the Great St. Petersburg Polytechnic University, 195251 St. Petersburg, Russia; veronika.prihodko@pharminnotech.com (V.P.); dasha0703@mail.ru (D.C.); susoyev92@mail.ru (Y.S.); quakenbush97@gmail.com (N.Z.); 2Department of Pharmacology and Clinical Pharmacology, Saint Petersburg State Chemical Pharmaceutical University, 197022 St. Petersburg, Russia; okovityy@mail.ru; 3N.P. Bechtereva Institute of the Human Brain of the Russian Academy of Sciences, 197376 St. Petersburg, Russia; 4Institute of Translational Biomedicine, Saint Petersburg State University, 199034 St. Petersburg, Russia

**Keywords:** TRPC6, Alzheimer’s disease, cerebral ischemia, pharmaceutical agents

## Abstract

Alzheimer’s disease and cerebral ischemia are among the many causative neurodegenerative diseases that lead to disabilities in the middle-aged and elderly population. There are no effective disease-preventing therapies for these pathologies. Recent in vitro and in vivo studies have revealed the TRPC6 channel to be a promising molecular target for the development of neuroprotective agents. TRPC6 channel is a non-selective cation plasma membrane channel that is permeable to Ca^2+^. Its Ca^2+^-dependent pharmacological effect is associated with the stabilization and protection of excitatory synapses. Downregulation as well as upregulation of TRPC6 channel functions have been observed in Alzheimer’s disease and brain ischemia models. Thus, in order to protect neurons from Alzheimer’s disease and cerebral ischemia, proper TRPC6 channels modulators have to be used. TRPC6 channels modulators are an emerging research field. New chemical structures modulating the activity of TRPC6 channels are being currently discovered. The recent publication of the cryo-EM structure of TRPC6 channels should speed up the discovery process even more. This review summarizes the currently available information about potential drug candidates that may be used as basic structures to develop selective, highly potent TRPC6 channel modulators to treat neurodegenerative disorders, such as Alzheimer’s disease and cerebral ischemia.

## 1. Introduction

Due to increased life expectancies, neurodegenerative diseases (NDD), such as Alzheimer’s disease (AD), dementia, and cerebrovascular diseases, are considered by WHO the main cause of disability in the coming decades. Currently, there are no effective disease-modifying or preventing therapies for those NDDs.

AD is caused by the progressive loss of neurons in brain structures that are responsible for memory acquisition and preservation, such as the hippocampus and cortical areas. The exact mechanism that causes neurons to die is not known. Among the most studied toxic changes of proteins that cause neuronal degeneration in AD are extracellular amyloid-beta (Aβ) aggregates and intracellular hyperphosphorylated tau (p-tau) that forms neurofibrillary tangles. Recently, disruption of immune system signaling in glial cells has started to receive growing attention due to the appearance of genome-wide association study (GWAS) data for late-onset AD patients [1].

Brain ischemia is a cerebrovascular disease that is caused by a restriction in blood supply, leading to oxygen deprivation and the rapid death of neurons. Several mechanisms, including excitotoxicity, ionic imbalance, oxidative and nitrosative stress, and apoptosis, have been implicated in ischemic neuronal death [2,3]. Acute brain ischemia can be treated successfully in modern healthcare settings, although treatment success depends on how quickly the patient receives medical care as well as the brain volume affected. However, there are currently no effective treatment options for chronic brain ischemia, which is usually caused by cerebral atherosclerosis.

Although the pathophysiological mechanisms causing AD and cerebral ischemia may differ, cerebral ischemia serves as a risk factor for AD development [4], and vice versa [5,6], indicating that a common intracellular mechanism may be disrupted in these two distinct pathologies. Such a common mechanism may be associated with Ca^2+^ dyshomeostasis. The N-methyl-D-aspartate (NMDA) receptor, an important excitatory neurotransmitter receptor, has been reported as the key player in the Ca^2+^ signaling in AD [7] and cerebral ischemia [8]. However, the NMDA receptor blocker memantine only relieves the symptoms temporarily in the early to moderate stage of AD patients [9,10]. Moreover, blocking NMDA receptors to prevent ischemic neuronal damage in clinical trials has caused severe adverse effects [11,12]. Although multiple factors might have contributed to the unsuccessful clinical trials, it is possible that the disruption of neuroprotective pathways, which precedes NMDA receptors hyperactivation, could be responsible for either AD and/or ischemic brain damage.

One of such neuroprotective pathways is the transient receptor potential cation channel, subfamily C, member 6 (TRPC6)-dependent regulation of excitatory synapse formation. TRPC6 overexpression has been shown to increase dendritic spine density [13] and rescue mushroom spine loss in mouse AD models [14], as well as protect neurons from ischemic brain damage [15,16]. TRPC6 upregulates the cAMP-response element-binding protein (CREB) pathway that is important for dendritic growth [17] and promotes synapse and dendritic spine formation, spatial memory, and learning [13]. In addition, TRPC6 also acts as a negative regulator that suppresses NMDA-induced Ca^2+^ influx in hippocampal neurons [15], which may protect neurons from excitotoxicity in the first stages of the disease. TRPC6 overexpression has been observed in breast cancer cells [18]. Overactivation of TRPC6 is toxic to immune cells (reviewed here [19]). Gain-of-function mutations of TRPC6 have been associated with familial forms of focal segmental glomerular sclerosis [20]. The mentioned studies indicate that despite the positive effect of TRPC6 activation in the brain, excessive TRPC6 activation has toxic effects on other cellular systems in the body.

Current prevalent evidence suggests that the TRPC6 channel function is downregulated in AD and cerebral ischemia [14,21,22,23]. However, there are reports in the literature that observe TRPC6 overactivation in AD [24] and cerebral ischemia models [25,26]. Existing contradictions on TRPC6 channel function in AD and ischemia indicate that these NDDs might be heterogenic, meaning that in one group of patients, the disease leads to hypofunction of the TRPC6 channel; however, there is another group (most likely smaller than the first one) where TRPC6 is hyperactivated. Thus, in order to preserve brain function, TRPC6 activity has to be in its physiological state, and deviations towards hypoactivity as well as hyperactivity are toxic to the cells. In terms of patient treatment, a careful investigation of the dysfunction of the TRPC6-dependent molecular pathway has to be performed in order to develop appropriate pharmacological treatments for said pathologies.

This review is devoted to the description of the role of TRPC6 channels in AD and brain ischemia with a particular focus on the dysfunction of them as Ca^2+^-dependent channels. Potential drug candidates that have shown their therapeutic effects in different cellular and animal models are discussed. When available, the pros and cons of each particular TRPC6 channel modulator are mentioned. TRPC6 is also involved in certain Ca^2+^-independent processes, such as amyloid precursor protein (APP) interaction [27]. However, in order to keep the review focused, this Ca^2+^-independent process observed in AD and brain ischemia pathogenesis is omitted.

## 2. TRPC Channels and Their Regulation in Cells

Transient receptor potential (TRP) cation channels form a large family of multifunctional cell sensors. There are 29 TRP channels that can be described and divided into six subfamilies based on sequence homology: seven canonical channels (TRPC), six vanilloid channels (TRPV), eight melastatin-related channels (TRPM), three polycystic channels (TRPP), three mucolipins (TPRML), and one ankyrin channel (TRPA) [28].

TRPCs participate in different physiological processes in the development of the nervous system [29]. TRPC subfamily consists of seven members: TRPC1, TRPC2, TRPC3, TRPC4, TRPC5, TRPC6, and TRPC7. TRPC2 is a pseudogene in a number of vertebrates, including humans [30].

Analysis of the expression of different types of TRPC channels in situ using the Allen Brain Atlas and of gene expression in different regions of the mouse brain has demonstrated that TRPC6 is highly expressed in the hippocampus [14]. Western blot analysis of the rat hippocampus has shown that TRPC6 levels are enhanced in postsynaptic structures compared with synaptosomes. Moreover, electron microscopy has shown that TRPC6 is mostly located in the postsynaptic sites [13].

TRPs are cloned and identified, assuming that they are calcium-selective and activated by the emptying of internal Ca^2+^ stores (store-operated channels (SOC)) [31]. After their initial functional characterization, it has turned out that both assumptions do not hold true, especially for TRPCs [31]. These ion channels only show a moderate Ca^2+^ selectivity (PCa/PNa from ~0.5 to 9), and the TRPC3/6/7 subfamily of TRPC channels can be activated by the second messenger diacylglycerol (DAG) produced by receptor-activated phospholipase C (PLC) without any involvement of internal stores (receptor-operated channels (ROC)) [32].

Store-operated Ca^2+^ entry (SOCE) via TRPCs happens when 1,4,5-trisphosphate (IP3) or some other intracellular mechanism empties Ca^2+^ stored in the endoplasmic reticulum (ER). The fall in the ER Ca^2+^ concentration signals to the plasma membrane to open store-operated channels [33]. The major breakthrough in the understanding of SOCE physiology happened when Ca^2+^-sensing stromal interaction molecule (STIM) ER proteins and plasma membrane (PM) Orai 1-3 channels have been described [34,35,36,37]. STIM1 and 2 proteins reside in the ER and monitor ER Ca^2+^ concentration with their EF-hand motif. When the ER Ca^2+^ concentration drops, Ca^2+^ dissociates from EF-hand, thus allowing STIM proteins to oligomerize and move to ER-PM tight junctions where they interact with PM Orai and TRP channels in order to facilitate Ca^2+^ flow.

## 3. Role of TRPC6 in the Formation of Excitatory Synapses

TRPC6 plays a certain physiological role in the formation of excitatory synapses. Particularly, TRPC6 overexpression increases dendritic spine density [13] and attenuates mushroom spine loss in presenilin 1 knock-in (PS1-KI) and amyloid precursor protein knock in (APP-KI) hippocampal neurons to the level of wild type neurons [14]. At the same time, overexpression of TRPC1, TRPC3-5, and TRPC7 does not affect spine morphology [14]. Meanwhile, the downregulation of TRPC6 expression leads to spine density reduction, lowers the frequency of spontaneous miniature excitatory postsynaptic currents (mEPSCs) [13], decreases the expression of postsynaptic density protein 95 (PSD95), and inhibits phosphorylation of calcium-calmodulin-dependent protein kinase II (pCaMKII) [14].

Activation of TRPC6 can promote spine formation via the STIM2-neuronal SOCE-CaMKII pathway. The expression level of STIM2 is downregulated in hippocampal neurons from PS1-M146V-KI [38] and APP-KI [14] mice as well as in a cellular model of amyloid synaptotoxicity [39]. The STIM2 reduction may be a compensatory response to ER Ca^2+^ overload in these models and may lead to the loss of mushroom spines in PS1-M146V-KI and APP-KI hippocampal neurons. TRPC6 and Orai2 channels are suggested to be the key components of neuronal SOCE in hippocampal cells [14]. It is hypothesized that the downstream molecule for TRPC6-mediated SOCE in hippocampal neurons is CaMKII since upregulation of neuronal SOCE activity recovers phosphorylation of CaMKII, restores the number of mushroom spines in hippocampal neurons in mouse models of AD [38,39], and induces long-term potentiation in PS1-KI, APP-KI, and 5xFAD hippocampal slices [14,21].

Alternative TRPC6-downstream signaling pathways are the Ca^2+^/calmodulin-dependent kinase IV (CaMKIV) pathway and the cAMP-response element-binding protein (CREB) pathway, which is important for dendritic growth in hippocampal neurons [17]; another study has shown that the CaMKIV-CREB pathway is important in promoting synapse and dendritic spine formation, spatial memory, and learning [13]. TRPC6 also acts as a negative regulator that suppresses NMDA-induced Ca^2+^ influx in hippocampal neurons [15]. In addition, NMDAR has been shown to regulate transcription and degradation of TRPC6 in neurons in a bidirectional manner through NMDAR subunit 2A (NR2A) or NR2B activation [40].

## 4. Hypo- and Hyperactivation of TRPC6 Channels in Different Pathogenetic Forms of AD

There is evidence that different genetically inherited familial forms of AD (fAD) can cause TRPC6 dysfunction [14,21,24]. Both hypo- [14,21,22] and hyperactivation [24] of TRPC6 channels have been reported for different fAD-associated mutations in *APP* and *PS* genes.

fAD-associated PS2-N141I, M239V mutations cause downregulation of TRPC6-mediated Ca^2+^ entry in transiently transfected HEK cells. Lessard et al. suggested that TRPC6 downregulation by PS2-N141I, M239V does not depend on ER Ca^2+^ content but rather involves the interaction of PS2 with an intermediate protein of unknown origin [22]. Later on, contradictory results were obtained on ER Ca^2+^ content in PS2-N141I expressing cells [41,42]. In double knockout fibroblasts, PS2-N141I increases ER Ca^2+^ content [41] but lowers it in Hela cells [42]. According to the classical understanding of the SOCE physiology, overloaded ER Ca^2+^ stores downregulate SOCE [33]; thus, the data provided by Tu et al. [41] seem to be more relevant.

PS1-M146V is an fAD-associated mutation that has been reported to downregulate TRPC6-dependent Ca^2+^ entry in hippocampal neurons in store-operated mode [14]. PS1-M146V has been shown to increase ER Ca^2+^ content in mouse embryonic fibroblasts (MEFs) [41] and in neurons [43]. It has been demonstrated in previous studies [38,44] that ER Ca^2+^ stores are overloaded in neurons from AD mouse models. Furthermore, it has been discovered that overloaded ER Ca^2+^ stores cause compensatory downregulation of TRPC6 channels in PS1-M146V neurons [14]. A similar impact on TRPC6 function has been reported for fAD-associated mutations in APP (KM670/671NL and I716F) [14] as well as for Aβ toxicity in a cell culture model of AD [21]. There are fAD mutations in PSEN1 (PSEN1M146L, PSEN1S170F, PSEN1I213F, PSEN1E318G, PSEN1P117R, PSEN1L226F, PSENA246E) [45,46] and PSEN2 (PSEN2M239I, PSEN2T122R) [42], which have been reported to downregulate SOCE, although their role in the regulation of TRPC6 function has not been investigated yet. TRPC6 activators have been shown to recover the percentage of mushroom spines in cell culture models of fAD and induce long-term potentiation in hippocampal brain slices taken from AD mouse models [14,21]. Based on these results, it is suggested that activators of TRPC6 may have a therapeutic value for the treatment of fAD with TRPC6 hypofunction [14,47,48,49].

PS1-ΔE9 mutation has been reported to empty ER Ca^2+^ stores [41] and enhance SOCE [24,50]. There are other fAD mutations in PSEN1 (PSEN1D257A, PSEN1D385A), which have been reported to enhance SOCE [51]; however, their role in the regulation of TRPC6 function has not been investigated yet. Today, there is only one fAD-associated PS1-ΔE9 mutation that has been shown to upregulate the TRPC6 function in store-operated mode [24]. A TRPC6 inhibitor has been shown to recover mushroom spine percentage in a cell culture model of fAD [24]. Inhibitors of TRPC6 have been proposed to have therapeutic effects in fAD with TRPC6 hyperfunction [24,49].

To conclude the section, in order to normalize TRPC6 function in neurons and preserve the stability of excitatory synaptic contacts, suitable pharmacological agents have to be used for distinct genetic forms of AD.

## 5. Cerebral Ischemia as a Risk Factor for AD Development

Recent experimental and clinical findings have demonstrated a high degree of correlation between cerebral ischemia and AD [4,52,53]. While some studies have indicated that ischemic stroke significantly increases the risk of AD [4], others, in turn, have associated AD with a higher risk of stroke [5,6]. Previous studies have suggested that almost 30% of AD subjects bear evidence of cerebral infarction at autopsy [54,55]. A meta-analysis that comprises seven cohort studies and two nested case-control studies has found that a history of stroke is associated with the development of AD [4]. Notably, several lines of evidence suggest that AD patients have a high risk of cerebral ischemia [5,6]. For example, among AD patients with no history of previous stroke, vascular dementia, or other cerebral degenerative diseases, the incidence of ischemic stroke amounts to 37.8 per 1000 persons (versus 23.2 in non-AD controls) [5]. In another register-based matched cohort study [6], patients with Alzheimer’s dementia have a higher risk of hemorrhagic stroke, while there is no difference in ischemic stroke incidence. When the results are analyzed within different age groups, the risk of ischemic stroke is found to be increased among AD patients younger than 80 years [6].

## 6. Role of TRPC6 in the Development of Ischemia

Ca^2+^ overload is one of the main molecular mechanisms involved in ischemic cell damage and death [56]. TRPC6, along with a few other prominent members of the family, has recently gained considerable attention as a promising target for the prevention of Ca^2+^ overload [23]. Dysregulation of TRPC6 activity has been implicated in ischemic stroke [23,57], as well as retinal ischemia [58], and renal hypoxia following cerebral ischemia [59].

On the one hand, upregulation and maintenance of TRPC6 activity prevent NMDAR hyperactivation and the subsequent Ca^2+^ influx, development of excitotoxicity, and neuronal death. In supporting this notion, both direct activation by 1-oleoyl-2-acetyl-sn-glycerol (a synthetic analog of diacylglycerol, the main endogenous TRPC6 agonist) and overexpression of TRPC6 inhibit NMDA-induced currents in cultured hippocampal neurons [15], and TRPC overexpression attenuates excitotoxic damage in hippocampal and cortical neurons [16]. *Trpc6*-transgenic mice with an elevated basal level of TRPC6 expression are less susceptible to cerebral ischemia than their wild-type littermates and have lower mortality rates, reduced infarct volumes, and better neurological outcomes after middle cerebral artery occlusion (MCAO) [16]. Recent studies have shown that TRPC6-mediated signaling promotes neuronal survival [60], the brain-derived neurotrophic factor-mediated axonal growth cone guidance [61], dendritic outgrowth and branching [17], and excitatory synapse formation [13]. In addition, positive modulation of TRPC6 activity allows for the sustained activation of the CREB/CaMK-IV and Ras/MEK/ERK pathways, which is vital for neuronal development, survival, and proper functioning [23]. Blocking CREB signaling hinders post-stroke recovery [62], and CaMK-IV inhibition impairs blood–brain barrier integrity and exacerbates ischemic injury [63]. In turn, elevated CREB and CaMK-IV activity is associated with improved post-stroke outcomes in a number of animal studies [62,63,64,65,66].

On the other hand, some experimental data suggest that the upregulation of TRPC6 activity increases intracellular Ca^2+^ concentrations concomitantly with NMDAR activation, further exacerbating excitotoxic damage to neurons [16]. Oxygen-glucose deprivation in cultured cortical neurons and MCAO in wild-type mice are associated with elevated TRPC6 expression and activity, while TRPC6 deletion attenuates glutamate- and NMDA-induced cytotoxicity and reduced infarct volumes [25]. Knockdown of TRPC3, 6, and 7 prevents apoptosis in cultured astrocytes and ameliorates ischemic brain injury in mice [26]. Moreover, prevention of TRPC6 hyperactivation results in increased neuronal viability, reduced infarct volumes and brain edema, and improved functional recovery following acute ischemic stroke in rats [67,68,69] and crab-eating macaques [67]. Other TRPC6 inhibitors have also been reported to exert beneficial effects in experimental models to some extent relevant to ischemic brain injury (e.g., acute renal ischemia/reperfusion injury) [70,71,72,73,74].

Existing controversies regarding the TRPC6 function in the development of brain ischemia might be due to different experimental settings (i.e., rodent model, sex, age, the method used to model ischemia). However, similarly to AD, brain ischemia seems to be heterogenic, meaning that one group of patients has TRPC6 hypofunction, and another one has TRPC6 hyperfunction. This indicates the need to develop pathology-dependent strategies to treat different NDD patients.

## 7. Available Drug Candidates to Modulate TRPC6 Activity

To date, two strategies have been proposed for the pharmacological modulation of TRPC6 activity for the treatment of Alzheimer’s disease and cerebral ischemia: (1) TRPC6 activation to allow Ca^2+^ influx via neuronal SOCE and sustain the stability of postsynaptic contacts (for AD) and to attenuate NMDAR activity and prevent calcium-dependent excitotoxicity [15,16] (for ischemia); (2) inhibition of TRPC6 in order to prevent calcium overload and the subsequent cell damage [23,64] (for AD and ischemia). Although apparently mutually exclusive, both of these strategies are aimed to keep the intracellular Ca^2+^ concentration within the normal range, which requires limiting its entry via transmembrane channels and/or release from intracellular stores [75]. Given the pivotal role of NMDAR in excitotoxic neuronal damage, NMDAR blockers have been proposed as potential neuroprotective agents, although most of them have failed to show substantial effectiveness in human patients so far [76]. Due to that fact, TRPC6 has emerged as an alternative therapeutic target for AD and ischemic stroke [23].

### 7.1. TRPC6 Activators

TRPC6 activation can be induced by several endogenous diacylglycerols (DAGs) [32], lysophosphatidylcholines [75], and 20-hydroxyeicosatetraenoic acid, which is a metabolite of arachidonic acid [77]. A number of DAG analogs, including 1,2-dioctanoyl-*sn*-glycerol, DAG-containing arachidonic and docosahexaenoic acids [78,79], and the docosanoid neuroprotectin D1 [80], have also been reported as TRPC6 agonists. This channel can also be activated by agents of synthetic or natural origin that are structurally different from DAG. Direct TRPC6 agonists acting in receptor-operated mode include synthetic compounds, such as flufenamic acid [81] and several pyrazolopyrimidine [82] and piperazine [83] derivatives. The benzimidazole-based small molecule agonist GSK1702934A, its azobenzene derivative OptoBI-1 [84], and the chromone-containing compound C20 [85] are also thought to be direct (ROC) stimulators of TRPC6 activity. In contrast, certain naturally occurring chemicals are known to potentiate TRPC6 effects in an indirect manner. These include the stilbenoid resveratrol [66], the isoflavone calycosin [86], and (−)-epigallocatechin-3-gallate, a catechin-type polyphenol [87]. Recently, a novel potent ethanolamine derivative, bis-{2-[(2E)-4-hydroxy-4-oxobut-2-enoyloxy]-N,N-diethylethanaminium} butandioate (FDES), has been demonstrated to exert neuroprotective effects due to selective TRPC6 channel activation in store-operated mode [88,89]. The aminoquinazoline derivative, NSN21778, was demonstrated by Zhang et al. to have a store-dependent mechanism of action where DAG is required as a co-factor for TRPC6 activation [14]. The piperazine derivative, 51164, has been shown to activate TRPC6 in store-operated mode requiring DAG as a co-factor [21].

#### 7.1.1. Endogenous Ligands and Analogs

##### DAG

A compound containing both DAG and arachidonic acid fragments, 1-stearoyl-2-arachidonyl-sn-glycerol (SAG), elicits a rapid Ca^2+^ flux into HEK293 cells, while 1-stearoyl-2-docosahexaenoyl-sn-glycerol (SDG), which does not contain an arachidonic acid moiety, has significantly lower potency [78]. 1-oleoyl-2-acetyl-*sn*-glycerol (OAG), a diacylglycerol analog and a TRPC3/6/7 channel modulator, is found to cross the plasma membrane and intracellularly activate the channels [32,90]. OAG has been shown to activate Ca^2+^-permeable channels, displaying TRPC6-like properties in cultured cortical neurons [91]. In addition, OAG has been demonstrated to increase field excitatory postsynaptic potential (fEPSP) levels in a TRPC-dependent manner in hippocampal slices from wild-type mice [92], indicating that it might have neuroprotective effects. However, as far as we are aware, those compounds have not yet been evaluated in in vivo models of AD and cerebral ischemia.

##### Lysophosphatidylcholine

Lysophosphatidylcholine (LPC) is produced from phosphatidylcholines via partial hydrolysis generally catalyzed by phospholipase A2. Increased LPC production has been observed in various disorders of the central nervous system, including stroke and AD [93], and is associated with acute and chronic brain ischemia [94,95]. Results of a cohort study have suggested that LPC levels could be used as a tool for ischemic stroke risk stratification in patients who have suffered a transitory ischemic attack before [96]. LPC is shown to activate TRPC6 channels and promote Ca^2+^ flux into endothelial cells, hampering their migration and preventing endothelial healing, thus contributing to atherogenesis [97,98].

##### 20-Hydroxy-5*Z*,8*Z*,11*Z*,14*Z*-Eicosatetraenoic Acid

20-hydroxy-5*Z*,8*Z*,11*Z*,14*Z*-eicosatetraenoic acid (20-HETE) is the main eicosanoid metabolite of arachidonic acid and a potent inflammatory vasoconstrictor. In HEK293 cells, 20-HETE (half maximal effective concentration, EC_50_ of 0.8 μM) elicits a three-fold increase in TRPC6 activity (as indicated by an increased inward, the non-selective current observed in whole-cell patch-clamp recordings) but does not affect intracellular Ca^2+^ concentrations [77]. In isolated guinea pig airway smooth muscle cells, it induces a dose-dependent inotropic effect via TRPC6 activation and the subsequent promotion of Ca^2+^ entry [99]. Nevertheless, 20-HETE has been shown to have detrimental effects in ischemic and traumatic brain injury, which might be explained by its vasoconstrictor properties, and has even been proposed as a predictor of poor prognosis in stroke patients [100]. To our knowledge, 20-HETE has not yet been tested in AD models, but it has been shown to activate TRPV1 channels in dorsal root ganglia cultures [101].

##### 10*R*,17*R*-Dihydroxydocosa-4*Z*,7*Z*,11*E*,13*E*,15*Z*,19*Z*-Hexaenoic Acid

10*R*,17*R*-dihydroxydocosa-4*Z*,7*Z*,11*E*,13*E*,15*Z*,19*Z*-hexaenoic acid (Neuroprotectin D1, NPD1, Table 1) is a docosahexaenoic acid ((4*Z*,7*Z*,10*Z*,13*Z*,16*Z*,19*Z*)-docosa-4,7,10,13,16,19-hexaenoic acid, DHA)-derived endogenous anti-inflammatory mediator commonly found in fish oil [102]. NPD1, among other neuroprotectins, is synthesized in ischemic brain tissue as a result of DHA enzymatic lipoxygenation. In a rat model of ischemic stroke, NPD1 administration is associated with a significantly elevated TRPC6 and CREB activity, while the inhibition of the MEK/ERK pathway results in a decrease in NPD1 neuroprotective activity. Continuous intracerebroventricular administration of NPD1 over 10 min at 2 h after reperfusion sustains TRPC6/CREB activity, reduces infarct volumes, and promotes functional recovery [80]. The role of NPD1 in TRPC6-mediated neuroprotection in AD has not been described yet.

#### 7.1.2. Hyperforin and Other Phytochemicals

##### Hyperforin

((1*R*,5*S*,6*R*,7*S*)-4-hydroxy-6-methyl-1,3,7-tris(3-methylbut-2-en-1-yl)-6-(4-methylpent-3-en-1-yl)-5-(2-methylpropanoyl)bicyclo[3.3.1]non-3-ene-2,9-dione) (Hyperforin, Table 1) is a phloroglucinol derivative and a major active constituent of St. John’s wort (*Hypericum perforatum* L.). Hyperforin is a potent inhibitor of TRPC6 proteolysis and a positive modulator of TRPC6/CREB activity, acting in a manner similar to that of the brain-derived neurotrophic factor (BDNF) [110,127]. It is thought to bind to TRPC6 due to structural similarities to DAG and has higher selectivity because of the relative rigidity of the phloroglucinol pharmacophore moiety [128]. Neuroprotective and antidepressant-like properties of hyperforin and hyperforin-containing *H. perforatum* preparations involve the modulation of axonal growth, neurite growth and branching, dendritic spine formation, and the promotion of neuronal plasticity [106,127]. As proposed by Singer et al., the increase in Na^+^ concentration resulting from TRPC6 activation by hyperforin might inhibit serotonin reuptake via the serotonin/Na^+^ symporter, which, together with increased synaptic plasticity, could explain the antidepressant-like activity of *H. perforatum*-based drugs [129]. Confirming this hypothesis, larixyl acetate, a selective blocker of TRPC6, abolishes the antidepressant-like effects of hyperforin observed in mice in the tail suspension test [130].

In an ex vivo experiment, hyperforin (0.3 μM) promotes mature stubby spine formation and decreases the proportion of immature thin spine formation in rat hippocampal pyramidal neurons but does not affect mushroom spine density and morphology. Proper TRPC6 expression level and the presence of a fully functional TRPC6 channel are required for hyperforin to exert its effects, which suggests the key role of TRPC6 activation in its mechanism of action [106]. Using different rodent models of ischemic stroke, hyperforin has been shown to promote post-stroke neuro- and angiogenesis [103,109], inhibit microglial activation [108], attenuate brain edema [104], stimulate hippocampal neurogenesis, ameliorate post-stroke depression and anxiety, and restore memory function [107]. Chronic hyperforin treatment stimulates the expression of the tropomyosine receptor kinase B (TrkB) BDNF receptors as well as of TRPC6 in murine cortical neurons but has no effect on hippocampal neurogenesis [131]. When applied intracerebroventricularly to rats immediately after MCAO, hyperforin preserves TRPC6 activity, reduces infarct volumes, promotes functional recovery, and increases neurologic scores at 24 h after reperfusion [110].

There is a lot of evidence that hyperforin and its derivatives are highly selective towards the TRPC6 channel and do not exert similar effects on its closest relative, the TRPC3 channel [106,127,128,132]. Several studies have shown that hyperforin activates TRPC6 and increases its expression [106,133], leading to a decrease in the Aβ level and an improvement in cognitive performance in AD models [92,105,111].

The neuroprotective effect of hyperforin has been demonstrated in several rodent models of AD. In rats co-injected with amyloid fibrils and hyperforin in the hippocampus, hyperforin reduces amyloid deposit formation and, therefore, decreases the Aβ-induced neurotoxicity, reactive oxidative species formation, and attenuated behavioral impairments [105]. A more stable hyperforin derivative, tetrahydrohyperforin, also prevents the cognitive decline and synaptic impairment in double transgenic APPswe/PSEN1ΔE9 mice in a dose-dependent manner. It has been shown that the neuroprotective mechanism of tetrahydrohyperforin is associated with a reduced rate of proteolytic processing of APP, decreased the total amount of fibrillar and oligomeric forms of Aβ, reduced level of tau hyperphosphorylation, and attenuated astrogliosis [132]. Tetrahydrohyperforin has been shown to specifically target TRPC6 [92].

Hyperforin is also thought to be responsible for the induction of the cytochrome P450 enzyme CYP3A4 by binding to and activating the pregnane X receptor [134], indicating that it might have side effects and undesirable drug-to-drug interactions. Moreover, hyperforin is difficult to synthetize [135], unstable when exposed to light, and irritant to the gastrointestinal tract [136]. Such side effects might limit the use of hyperforin as a TRPC6 activator.

##### Resveratrol

Resveratrol (5-[(*E*)-2-(4-hydroxyphenyl)ethenyl]benzene-1,3-diol) (Table 1), which is found in grapes, berries, peanuts, and other plants, is among the best-known natural compounds with antioxidant and antihypoxic activity [66,137,138]. In the pioneering study of Wang et al. [116], resveratrol is shown to be able to cross the blood–brain barrier (BBB) and exert neuroprotective activity. Injected intraperitoneally either during or shortly after the induction of cerebral ischemia, it largely prevents delayed neuronal cell death and glial cell activation in a gerbil model of transient bilateral common carotid artery occlusion [116]. Based on the observed increase in TRPC6 and CREB activity, which has been prevented by PD98059 or KN62, inhibitors of MEK and CAMKIV/ CaMKII, respectively, it is suggested that resveratrol has exerted its effects via TRPC6 activation. Since its administration has been accompanied by a marked decrease in calpain activity, resveratrol is classified as an indirect positive modulator of TRPC6 activity [66]. When given to rats for 7 days before MCAO, resveratrol significantly reduces infarct volumes and enhances neurological scores at 24 h after reperfusion [66]. Low-dose oral resveratrol treatment for three consecutive days before and after an ischemic stroke induced in rats by middle cerebral artery clipping alleviates brain damage caused by the following recurrent stroke. Resveratrol normalizes BBB function and reduces cerebral edema without affecting regional cerebral blood flow and systemic blood pressure [112]. Resveratrol preconditioning (48 h before the induction of ischemia) effectively prevents neuronal cell loss in a mouse MCAO model of stroke [113]. In a rat model of global cerebral ischemia induced by asphyxial cardiac arrest, resveratrol (48 h before the induction of ischemia) is shown to protect the CA1 region of the hippocampus similarly to ischemic preconditioning [115].

Resveratrol is actively used in AD research [139,140,141]. Neuroprotective effects of resveratrol have been shown to be associated with the activation of silent mating type information regulation 2 homolog 1 (SIRT1) and vitagenes production [142]. In a phase II trial in AD patients, resveratrol is safe and well-tolerated, but its effectiveness is contradictory [143,144].

Although generally considered as a non-toxic therapeutic agent, high doses of resveratrol inhibit CYP3A4 activity in vitro [145] and in healthy volunteers [146], thus potentially inhibiting drug metabolic clearance, increasing bioavailability and toxicity of drugs taken concomitantly [147].

#### 7.1.3. Synthetic Compounds

##### Flufenamic Acid

Flufenamic (2-[[3-(trifluoromethyl)phenyl]amino]benzoic) (Table 1) acid (FFA) is a member of the fenamate class of nonsteroidal anti-inflammatory drugs that have limited clinical applications due to their toxicity. FFA has been found to selectively activate TRPC6, at the same time, inhibiting TRPC3, 4, 5, and 7 [81]. Direct activation of TRPC6 by FFA has been confirmed in a number of in vitro studies [148], including those in glomerular podocytes [149] and ventricular cardiomyocytes [150]. Male (but not female) embryonic mice cortical neurons, which have been pre-treated with FFA 15 min before oxygen-glucose deprivation, have shown significantly higher viability, although this effect is not linked by the authors to TRPC6 activation [118]. Similar results are obtained in a recent in vitro glutamate toxicity assay using isolated rat embryonic hippocampal neurons [119].

Multiple experimental evidences suggest that FFA is a broad spectrum ion channel modulator, with a preference for non-selective cation channels and chloride channels (reviewed here [81]). Its activity seems to be dose-dependent since FFA inhibits TRPC6 with a half maximal inhibitory concentration (IC_50_) of 17.1 μM [151] but activates the same channel at 100 μM [148]. TRPM8 is inhibited by 100 μM FFA but is slightly activated at higher concentrations [152]. A worse situation is reported for big calcium-activated potassium channels (BK_Ca)_ modulation since FFA activates the channel below 10 μM, inhibits the channel between 10 to 50 μM, and then activates the channel above 50 μM [153]. Such opposing effects on the same channels and a huge number of other ion channels that are modulated by FFA makes it an inappropriate drug for usage in humans.

##### Piperazines

Sawamura et al. discovered a group of piperazine-based potent TRPC3/6/7 agonists functioning in receptor-operated mode with varying selectivity for different channel subtypes [83]. Among that group, 2-[4-(2,3-dimethylphenyl)piperazin-1-yl]-N-(2-ethoxyphenyl)acetamide (PPZ2, Table 1) dose-dependently activates TRPC6 and TRPC6-like channels in HEK cells, vascular smooth muscle cells, and cultured rat cerebellar granule neurons. PPZ2 and PPZ1 ([4-(5-chloro-2-methylphenyl)piperazin-1-yl]-3-fluorophenylmethanone) (Table 1) promote neurite outgrowth in a manner similar to that of BDNF and provide protection against serum deprivation-induced neuronal death [83].

Later on, another piperazine derivative, N-(2-chlorophenyl)-2-(4-phenylpiperazin-1-yl)acetamide (51164, Table 1), has been shown to activate the TRPC6 channel in store-operated mode with DAG acting as a co-factor [21]. Nanomolar concentrations of 51164 protect mushroom spines from amyloid toxicity, induce TRPC6-dependent neuronal SOCE in postsynaptic spines, and restore the induction of long-term potentiation in hippocampal slices taken from 6 months old 5xFAD mice [21].

*N*-[3-[4-[3-[bis(2-methylpropyl)amino]propyl]piperazin-1-yl]propyl]-1*H*-benzimidazol-2-amine (AZP2006), another piperazine derivative, attenuates Aβ and tau toxicity and improves cognitive performance in various mouse models [154]. Currently, AZP2006 is in phase 2 clinical trial in patients with progressive supranuclear palsy [155].

Piperazine derivatives as the majority of TRPC6 agonists cross-react with TRPC3 and TRPC7 [83], limiting their use as specific TRPC6 modulators. Among other side effects of piperazines is their hepatotoxicity [137]; however, hepatotoxicity has not been predicted by bioinformatical analyses for a 51164 compound [21]. Gastrointestinal hemorrhage and multiple organ failure have been predicted by bioinformatical analyses; thus, there is a need to search for the lowest therapeutic dose for the 51164 compound, and most likely, there is a need to modify its structure in order to minimize the mentioned side effects [21].

##### Bis-{2-[(2*E*)-4-Hydroxy-4-Oxobut-2-Enoyloxy]-*N*,*N*-Diethylethanaminium} Butandioate

Bis-{2-[(2*E*)-4-hydroxy-4-oxobut-2-enoyloxy]-*N*,*N*-diethylethanaminium} butandioate (Table 1), abbreviated as FDES, is an ethanolamine derivative known to possess antihypoxic, anti-ischemic, and neuroprotective properties [88,89,120,156]. Chronic FDES administration decreases mortality and improves motor function and coordination following permanent bilateral common carotid artery ligation [120] and reduces spatial memory deficit following middle cerebral artery (MCA) occlusion/reperfusion [89] in rats. Later, FDES was demonstrated to ameliorate fore- and hindlimb motor disturbances and increase overall locomotor activity in rats with unilateral traumatic brain injury [88].

FDES is shown to potentiate neuronal SOCE into postsynaptic spines in mouse hippocampal neurons [89]. Since TRPC6 knockdown abolishes the effects of FDES on neuronal SOCE (similarly to hyperforin), TRPC6 activation is suggested to be the primary mechanism of FDES neuroprotective action. Nanomolar concentrations of FDES effectively protect mushroom dendritic spines from amyloid synaptotoxicity, stabilizing and enhancing synaptic transmission, and preserving synaptic contact density [89]. Similarly to hyperforin, FDES decreases the proportion of immature thin and stubby spines in hippocampal neurons. When administered intraperitoneally to rats subjected to MCAO for 7 consecutive days starting from 1 h after reperfusion, FDES improves short-term spatial memory retention, as observed in the Barnes maze [89].

FDES is a precursor of choline and has been shown to have nootropic and actoprotective properties [89]. We assume that FDES would cross-react with muscarinic acetylcholine receptors, causing phospholipase C activation and production of IP3 and DAG. This cross-reactivity of FDES could further enhance its TRPC6-agonistic properties, although this hypothesis remains to be experimentally proven.

##### N-[4-[2-[(6-Aminoquinazolin-4-yl)Amino]ethyl]phenyl]acetamide

N-[4-[2-[(6-aminoquinazolin-4-yl)amino]ethyl]phenyl]acetamide (NSN21778, Table 1) was proposed by Zhang et al. as a novel positive modulator of the TRPC6/neuronal SOCE pathway acting in a manner similar to that of 51164. Despite its effectiveness in terms of improving mushroom spine morphology and TRPC6-mediated SOCE in PS1-KI and APP-KI hippocampal neurons and rescuing long-term potentiation in the APP-KI mouse model of AD, NSN21778 is found to have a rather poor pharmacokinetic profile and a low penetration of the blood–brain barrier [14].

#### 7.1.4. Other TRPC6 Agonists

Several compounds described below have been confirmed to activate the TRPC6 channel using in vitro assays. However, to the best of our knowledge, their specific neuroprotective properties remain unexplored. Given their ability to interact with TRPC6, these compounds can be considered as potential neuroprotective agents.

##### Pyrazolopyrimidines

A number of pyrazolopyrimidines obtained by Qu et al. are reported to be direct agonists of TRPC6, 3, and 7. Among the four pyrazolopyrimidines tested by Qu et al., ethyl 4-(7-hydroxy-2-methyl-3-(4-(trifluoromethyl)phenyl)-pyrazolo[1,5-a]pyrimidin-5-yl)piperidine-1-carboxylate (compound 4n) is most active towards TRPC6 (EC_50_ = 1.39 or 0.89 μM depending on the conditions). However, compound 4n has demonstrated a much higher affinity for TRPC3 and TRPC7 (EC_50_ = 0.019 and 0.090 μM, respectively) [82].

##### GSK1702934A and OptoBI-1

GSK1702934A (1,3-Dihydro-1-[1-[(5,6,7,8-tetrahydro-4H-cyclohepta[b]thien-2-yl)carbonyl]-4-piperidinyl]-2H-benzimidazol-2-one) has been reported by Xu et al. to activate TRPC3 and 6 (EC_50_ = 0.08 and 0.44 μM, respectively), acting directly and independent of protein lipase C signaling from the extracellular side [157]. OptoBI-1, an azobenzene moiety-containing photochromic derivative of GSK1702934A, is found to activate TRPC6 as well as TRPC3 and 7, although having a slightly higher affinity for TRPC3. Light treatment of cultured murine hippocampal neurons with OptoBI-1 suppresses action potential firing elicited by repetitive depolarizing current injections [84].

##### 3-(6,7-Dimethoxy-3,3-Dimethyl-3,4-Dihydroisoquinolin-1-yl)-2H-Chromen-2-One

Recently, a novel small-molecule allosteric TRPC6 agent, 3-(6,7-dimethoxy-3,3-dimethyl-3,4-dihydroisoquinolin-1-yl)-2H-chromen-2-one (C20), has been reported by Häfner et al. [85]. C20 (EC_50_ = 2.37 μM) selectively activates TRPC6 channels in several HEK cell lines while only slightly reducing the basal activity of TRPC3 and increasing that of TRPC7 and not affecting TRPC4 and 5 activity at all. Higher concentrations of C20 (10 μM) potentiate the efficacy of OAG and GSK1702934A, low-selective TRPC6 agonists described above, in HEK cells and freshly prepared human platelets [85].

It can be assumed that the mechanism of action of C20 involves TRPC6 sensitization and not with its activation per se; that is, C20 allows TRPC6 to be activated at a low basal concentration of DAG [85].

### 7.2. TRPC6 Inhibitors

AD seems to be a multifactorial disease with different pathogenic cascades occurring in different patients. In terms of TRPC6 channel dysfunction, there are forms of fAD, which demonstrate TRPC6 hyperfunction [24]. For those patients, TRPC6 inhibitors might be used in order to normalize intracellular Ca^2+^ homeostasis.

The pathogenesis of cerebral ischemia is not fully understood. Similarly to AD, cerebral ischemia involves several different pathological cascades. There are studies reporting that TRPC6-mediated Ca^2+^ and Na+ influx facilitates NMDAR activation and exacerbates excitotoxicity, while TRPC6 deletion attenuates neuronal damage and death following focal cerebral ischemia [25]. In such cases, where excessive TRPC6 activity seems to be present, the use of TRPC6 inhibitors might be beneficial.

Arachidonic acid, which is released following cerebral ischemia, can be metabolized to 20-hydroxyeicosatetraenoic acid (20-HETE). 20-HETE is a potent vasoconstrictor that may contribute to ischemic injury [67]. Synthetic 20-HETE has been shown to activate TRPC6 [77]. Thus, inhibition of 20-HETE production by HET0016 reduces TRPC6 activation [158]. Inhibition of 20-HETE synthesis by N-hydroxy-N’-(4-n-butyl-2-methylphenyl)formamidine (HET0016, Table 1) decreases infarct volumes and increases cortical cerebral blood flow in cerebral ischemia induced by asphyxia cardiac arrest in rat pups [69] and in transient MCAO-induced ischemia in adult rats [121]. In neonatal piglets subjected to 6 min of acute asphyxia, HET0016 potentiates the neuroprotective effects of delayed hypothermia, increasing neuronal viability, preventing seizure development, and reducing neurological deficit [122,123]. Another 20-HETE inhibitor known as N-(3-chloro-4-morpholin-4-yl)phenyl-N′-hydroxyimidoformamide (TS-011, Table 1) markedly decreases infarct volumes and improves functional recovery in rats and primates [67,68].

We are unaware of any direct investigations of neuroprotective effects of HET0016 and TS-011 (Table 1) in AD models. However, since patients with Alzheimer’s disease show an accumulation of (2E)-4-hydroxy-2-nonenal (HNE) adducts [159], and HET0016 is a potent inhibitor of ω- and ω-1-hydroxylation of HNE/HNA (4-hydroxynonanoic acid) [160], HET0016 could be of interest regarding its potential neuroprotective properties in AD.

In mice with closed-head traumatic brain injury, larixyl acetate, a naturally occurring diterpene, ameliorates endothelial dysfunction [74], which is closely associated with cerebral ischemia as well [161]. In isolated mouse lungs, larixyl acetate prevents the development of the acute hypoxic ventilatory response [72]. TRPC6 inhibition is now considered to be the primary mechanism of larixyl’s neuroprotective action [72,74].

Mefenamic (2-(2,3-dimethylphenyl)aminobenzoic acid) (MFA), meclofenamic (2-(2,6-dichloro-3-methylanilino)benzoic), and niflumic (2-{[3-(trifluoromethyl)phenyl]amino}pyridine-3-carboxylic) acids (Table 1), non-steroidal anti-inflammatory drugs structurally related to flufenamic acid, are potent inhibitors of TRPC6 and some other closely related ion channels [151]. These compounds have been shown to attenuate glutamate-evoked excitotoxicity in cultured rat embryonic hippocampal neurons similarly to flufenamic acid [119,124]. In a 3xTgAD mouse AD model, MFA ameliorates cognitive impairments [125].

Maier et al. discovered a novel TRPC3/6/7 inhibitor, the aminoindanol derivative 4-[[(1R,2R)-2-[(3R)-3-Amino-1-piperidinyl]-2,3-dihydro-1H-inden-1-yl]oxy]-3-chlorobenzonitrile dihydrochloride (SAR7334, Table 1), with a higher selectivity towards TRPC6 (IC_50_ of 7.9 nM, as indicated by patch-clamp data) [71]. Hou et al. found that TRPC6 knockout or inhibition by SAR7334 mitigates oxidative stress-induced apoptosis of renal proximal tubular cells, which is considered to play a major role in renal ischemia/reperfusion [73]. SAR7334 and the tryptoline derivative 8009-5364 are reported to diminish acute hypoxia-induced pulmonary vasoconstriction and pulmonary arterial pressure in isolated mouse lungs [70,71]. SAR7334 has no effect on SOCE in primary cortical neurons [126].

Some rare-earth metal ions, including La3+ and Gd3+, have been shown to inhibit TRPC6 activity [106,127,133]. But since those ions inhibit all channels of the TRP family (except TRPM2), they cannot be used as selective TRPC6 antagonists. Clotrimazole (1-[(2-chlorophenyl)diphenyl-methyl]-1*H*-imidazole) is an imidazole compound, which inhibits TRPM2, TRPM3, TRPV4, and TRPC6 channels [90]. In SOCE mechanism studies, 2-aminoethoxydiphenyl borate (2-APB) [90] and 1-[2-[3-(4-methoxyphenyl)propoxy]-2-(4-methoxyphenyl)ethyl]-1H-imidazole hydrochloride (SKF-96365) [162] are identified as agents targeting TRPC6. 2-APB blocks TRPC6 [163] and was later found to impact Orai channel functioning [164]. 2-APB has also been shown to block Ca^2+^ influx induced by acetylcholine or thapsigargin application but not by DAG [163]. SKF-96365 is considered to be an inhibitor of receptor- and store-operated elevation of intracellular calcium levels via entry through voltage-independent channels [162,165]. Studies have demonstrated successful blocking of TRPC6 using SKF-96365 (IC_50_ = 2 μM), but a less pronounced effect on TRPC3 is also observed (IC_50_ = 12 μM) [90]. 8009-5364 is another highly specific TRPC6 antagonist, which has an IC_50_ of 3.2 μM, and is considered a promising agent for the treatment of pulmonary hypertension [70]. Investigation of neuronal SOCE mechanism in striatal neurons has revealed 4-N-[2-(4-phenoxyphenyl)ethyl]quinazoline-4,6-diamine (EVP4593, Table 1) (IC_50_ = 300 nM [126]) to be an inhibitor of TRPC1 channels [166]. EVP4593 has also been found to target heteromeric but not homomeric TRCP1 channels [166]. Later on, EVP4593 was demonstrated to block Orai channels at 300 nM concentration [126]. In cultured hippocampal neurons exhibiting PSEN1ΔE9 mutation, TRPC6 hyperactivation is blocked by 30 nM EVP4593 [24]. EVP4593 has been shown to inhibit the nuclear factor kappa-b (NF-Kb) [167]. It has been also observed that NF-Kb downregulates TRPC6 protein expression [168]; thus, EVP4593-mediated inhibition of the NF-Kb pathway might cause an increase in TRPC6 protein expression. Whether this increase in TRPC6 protein expression would compete with EVP4593-mediated blockade of TRPC6-dependent SOCE is an open question.

## 8. Conclusions

The present review summarizes current knowledge on the role of TRPC6 channels in the development of two neurological disorders: Alzheimer’s disease and cerebral ischemia. Cerebral ischemia serves as a risk factor for AD, and vice versa. It is becoming evident that both diseases can be caused by either upregulation or downregulation of TRPC6 channels. Thus, understanding the nature of the disruption of this molecular pathway in each particular patient is extremely important for appropriate drug prescription.

TRPC6 is structurally similar to TRPC3 and 7, and therefore the majority of compounds do not act selectively on either of these three isoforms, making it difficult to develop selective TRPC6 pharmacological modulators. Moreover, a number of TRPC6 channel modulators are cross-reactive to other cellular targets, thus limiting their pharmacological potential. Toxicity profiles of some known TRPC6 modulators (such as FFA) require further structural optimization. The presence of a wide range of different chemical substances that are known to interact with TRPC6 channels as well as the availability of cryo-electron microscopy structures of TRPC6 and 3 [169] may allow determining the pharmacophore in order to design selective TRPC6 activators and inhibitors in the future. In turn, the development of selective TRPC6 channel modulators could help slow down the progression of AD, cerebral ischemia, and, most likely, other TRPC6-dependent diseases.

## Figures and Tables

**Table 1 cells-09-02351-t001:** Pharmacological modulators of TRPC6 in cerebral ischemia and AD models.

No.	Compound Name and Structural Formula	Experimental Model	Mode of Administration	Effect(s)	Reference(s)
**TRPC6 Activators**					
Endogenous ligands					
1	Neuroprotectin D1 10*R*,17*R*-Dihydroxydocosa-4*Z*,7*Z*,11*E*,13*E*,15*Z*,19*Z*-hexaenoic acid 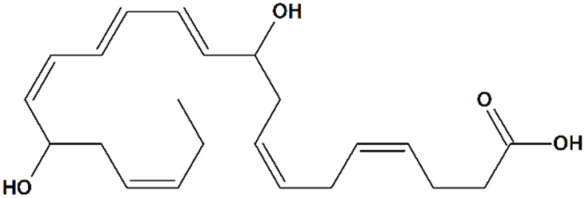	Rat transient MCAO	Icv injection at 2 h after reperfusion	reduced infarct volume reduced sensory and motor deficits	[80]
Phytochemicals					
2	Hyperforin (1*R*,5*S*,6*R*,7*S*)-4-Hydroxy-6-methyl-1,3,7-tris(3-methylbut-2-en-1-yl)-6-(4-methylpent-3-en-1-yl)-5-(2-methylpropanoyl) bicyclo[3.3.1]non-3-ene-2,9-dione 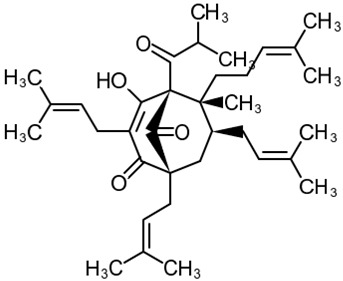	Lipopolysaccharide stimulation in 28 d post-MCAO isolated mouse astrocytes	Co-incubation for 16 h	increased viability	[103]
		Lipopolysaccharide stimulation in 28 d post-MCAO isolated mouse cortical neurons		increased viability	[103]
		NMDA toxicity in rat hippocampal slices	Co-incubation for 30 min	reduced edema	[104]
		Male Sprague-Dawley rats injected with fibrillary Aβ	Intrahippocampal co-injection of Aβ with the drug for 14 days	amyloid deposits disaggregation reduced spatial memory deficit	[105]
		Rat hippocampal slice cultures	Co-incubation for 24 h	increased proportion of mature stubby spines	[106]
		Hippocampal cultures from PS1-M146VKI and APPKI transgenic mice	Incubation for 16 h	increased percentage of mushroom spines in TRPC6-dependent manner increased neuronal SOCE in postsynaptic spines	[14]
		Hippocampal cultures treated with synthetic Aβ42 peptides	Co-incubation for 16 h	increased percentage of mushroom spines increased neuronal SOCE in postsynaptic spines	[21]
		Mouse transient MCAO	Intranasal administration q.d. for 7 d starting at day 7 post-MCAO	increased hippocampal neurogenesis improved post-stroke depression and anxiety reduced memory deficit	[107]
			Icv injections at 1, 24, and 48 h after MCAO	reduced microglial activation reduced infarct volume reduced neurological deficit	[108]
			Icv injections q.d. for 14 d starting at day 14 post-MCAO	increased angiogenesis reduced motor deficit	[109]
				increased angiogenesis increased subventricular neurogenesis reduced motor deficit	[103]
		Mouse permanent MCAO	Ip injection before ischemia onset	no effect on infarct volume or brain edema	[104]
		Mouse water intoxication			
		Rat transient MCAO	Icv injection at 6, 12, or 24 h after reperfusion	prevented neuronal apoptosis reduced infarct volume reduced neurological deficit	[110]
3	Tetrahydrohyperforin (1S,5S,7S,8R)-4,9-dihydroxy-1-(1-hydroxy-2-methylpropyl)-8-methyl-3,5,7-tris(3-methylbut-2-enyl)-8-(4-methylpent-3-enyl)bicyclo[3.3.1]non-3-en-2-one 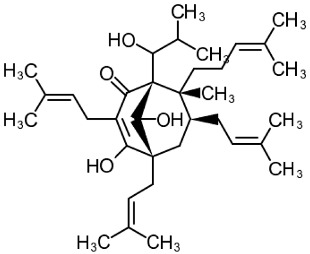	APPSEN1ΔE9 mice	Ip injections for 4 weeks	reduced memory deficit reduced amyloid deposition attenuated neuroinflammation and oxidative stress	[111]
4	Resveratrol 5-[(*E*)-2-(4-Hydroxyphenyl)ethenyl] benzene-1,3-diol 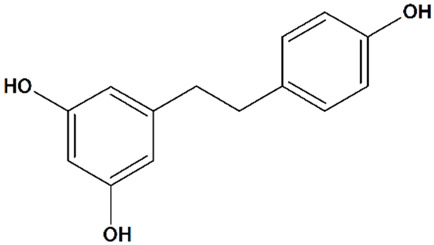	Oxygen/glucose deprivation in isolated rat brain endothelial cells	Preincubation for 3 d	increased viability	[112]
		Mouse transient MCAO	Ip injection at 48 h before MCAO	reduced infarct volume no effect on cerebral blood flow	[113]
			Oral gavage q.d. for 7 d starting from 24 h after MCAO, or for 5 d starting from 72 h after MCAO	increased vascular density in the basal ganglia region and cortex reduced infarct volume reduced neurological deficit	[114]
		Rat transient MCAO	Ip injections q.d. for 7 d before MCAO	reduced infarct volume reduced neurological deficit	[66]
		Rat recurrent transient MCAO	Oral gavage q.d. for 3 d between strokes	reduced infarct volume following an initial and recurrent stroke reduced blood–brain barrier disruption following a recurrent stroke reduced brain edema following a recurrent stroke reduced astrogliosis following a recurrent stroke no effect on cerebral blood flow during or after recurrent stroke	[112]
		Rat asphyxial cardiac arrest	Ip injection at 48 h before cardiac arrest	enhanced ATP synthesis efficiency in hippocampal mitochondria prevented hippocampal neuronal apoptosis	[115]
		Gerbil transient bilateral common carotid artery occlusion	Ip injections during occlusion or at reperfusion + at 24 h after reperfusion	reduced hippocampal microglial activation prevented hippocampal-delayed neuronal death	[116]
		Clinical trials in patients	diverse	affected neuroinflammation, Aβ deposition, and adaptive immunity in patients with mild to moderate Alzheimer’s disease	for a review, see [117]
Synthetic compounds					
5	Flufenamic acid 2-[[3-(Trifluoromethyl)phenyl]amino]benzoic acid 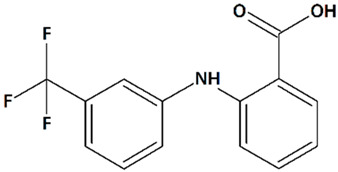	Oxygen/glucose deprivation in isolated mouse embryonic cortical neurons	Incubation from 15 min before oxygen/glucose deprivation until 24 h of reoxygenation	increased viability of male neurons no effect on female neurons	[118]
		Glutamate toxicity in isolated rat embryonic hippocampal neurons	Co-incubation for 10 min	increased viability	[119]
6	PPZ1 [4-(5-Chloro-2-methylphenyl)piperazin-1-yl]-3-fluorophenylmethanone 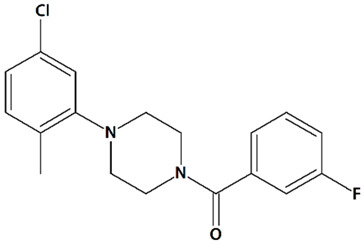	Serum deprivation in isolated rat cerebellar granule neurons	Incubation for 24 h before and 24 h after serum deprivation	increased neurite outgrowth increased cell viability	[83]
7	PPZ2 2-[4-(2,3-Dimethylphenyl)piperazin-1-yl]-N-(2-ethoxyphenyl)acetamide 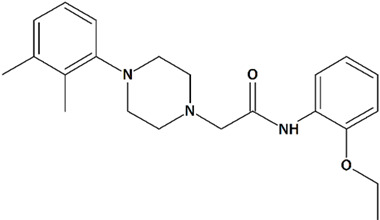				
8	51164 N-(2-chlorophenyl)-2-(4-phenylpiperazin-1-yl)acetamide 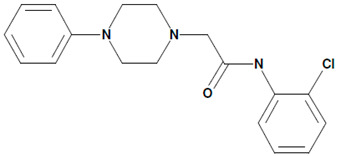	Aβ42-induced toxicity in primary hippocampal neurons	Co-incubation for 16 h	restored mushroom spines percentage induced neuronal SOCE in postsynaptic spines	[21]
		6 month-old 5xFAD mouse hippocampal slices	30 min incubation	restored LTP induction	
9	FDES Bis-{2-[(2E)-4-hydroxy-4-oxobut-2-enoyloxy]-N,N-diethylethanaminium} butandioate 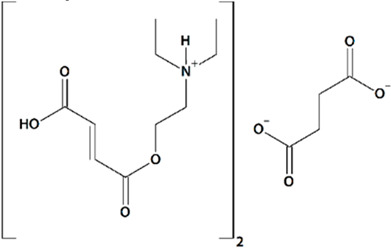	Aβ42-induced toxicity in primary hippocampal neurons	Incubation for 16 h	restored mushroom spines percentage induced neuronal SOCE in postsynaptic spines	[89]
		Rat transient MCAO	Ip injections at 1 h after reperfusion + q.d. for 7 d after reperfusion	improved spatial memory retention no effect on mortality	
		Rat permanent bilateral common carotid artery ligation	Oral gavage at 30 min before MCAO + q.d. for 21 d after reperfusion	reduced mortality reduced motor deficit reduced aggressiveness reduced emotional lability increased exploratory behavior	[120]
10	NSN21778 N-[4-[2-[(6-aminoquinazolin-4-yl)amino]ethyl]phenyl]acetamide 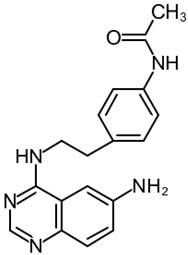	Primary hippocampal cell culture models from PS1-M146V and APPKI mice	Incubation for 16 h	increased percentage of mushroom spines in TRPC6-dependent manner increased neuronal SOCE in postsynaptic spines	[14]
		Hippocampal brain slices from PS1-M146V and APPKI mice	Pretreatment for 30 min	recovered LTP induction	
**TRPC6 inhibitors**					
11	HET0016 N-Hydroxy-N′-(4-n-butyl-2-methylphenyl)formamidine 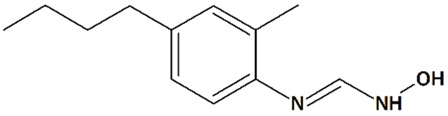	Rat pediatric asphyxial cardiac arrest	Iv injection at reperfusion	increased cortical CBF reduced brain edema	[69]
			Iv injection at reperfusion + ip injections every 6 h for 24 h after reperfusion	reduced neurological deficit reduced neurodegeneration	
		Rat transient MCAO	Iv injection immediately before reperfusion	reduced infarct volume	[68]
			Ip injections q.d. for 3 d before and 3 d after MCAO	increased CBF reduced infarct volume	[121]
		Piglet neonatal hypoxia/ ischemia	5 min-infusion at 5 min after reperfusion + hypothermia at 3 h after reperfusion	increased neuronal viability in the putamen, cortex, and thalamus prevention of seizures	[122]
			Iv injection at 5 min after reperfusion	increased neuronal viability in the putamen reduced neurological deficit no effect on cerebral blood flow (CBF)	[123]
12	TS-011 N-(3-Chloro-4-morpholin-4-yl)phenyl-N′-hydroxyimidoformamide 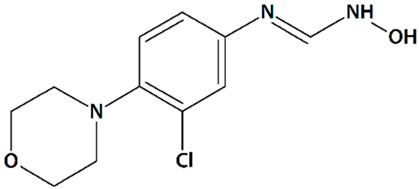	Rat transient MCAO	Iv injection at 30 min before MCAO + 1 or 2 h-infusion during MCAO	reduced cortical, subcortical, and total infarct volumes reduced the delayed drop in CBF no effect on volume at risk	[68]
			Iv injection at 20 min after MCAO + 2 h-infusion at reperfusion	reduced cortical and total infarct volumes no effect on volume at risk no effect on CBF	
			1 h-infusion at reperfusion, 1, 2, or 4 h after reperfusion	reduced infarct volumes	[67]
			1 h-infusion at reperfusion + iv injections q.d. for 7 d	reduced infarct volumes reduced sensory and motor deficits	
		Crab-eating macaque thrombotic internal carotid artery occlusion	Iv injection + 24 h-infusion after embolization	reduced infarct volume (when co-administered with tissue plasminogen activator) reduced neurological deficit	
13	Mefenamic acid 2-(2,3-Dimethylphenyl)aminobenzoic acid 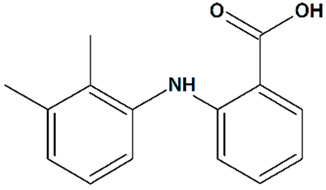	Glutamate toxicity in isolated rat embryonic hippocampal neurons	Co-incubation for 10 min	increased cell viability	[124]
		3xTg mice AD model	Administration by osmotic minipump over 28 days	reduced cognitive deficit	[125]
		Rat transient MCAO	Iv injection before MCAO	no effect on infarct and penumbra volumes and brain edema	[124]
			Iv injections at 1 h before + at 1, 2, and 3 h after MCAO	reduced infarct volume reduced brain edema	
			Icv 24 h-infusion starting at 1 h before MCAO		[119]
14	Meclofenamic acid 2-(2,6-Dichloro-3-methylanilino)benzoic acid 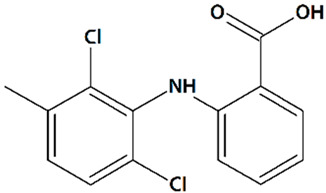	Glutamate toxicity in isolated rat embryonic hippocampal neurons	Co-incubation for 10 min	increased viability	[119]
15	Niflumic acid 2-{[3-(Trifluoromethyl)phenyl]amino}pyridine-3-carboxylic acid 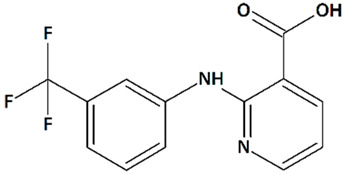				
16	SAR7334 4-[[(1R,2R)-2-[(3R)-3-Amino-1-piperidinyl]-2,3-dihydro-1H-inden-1-yl]oxy]-3-chlorobenzonitrile dihydrochloride 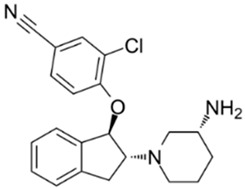	Primary cortical neurons	Treatment with 1 μM SAR7334 at the time of imaging	no effect on neuronal SOCE	[126]
17	EVP4593 4-N- [2- (4-phenoxyphenyl)ethyl]quinazoline-4,6-diamine 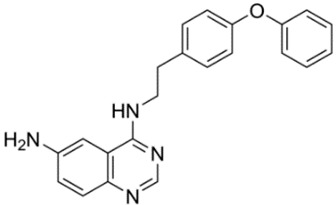	PSEN1ΔE9-hyperexpressing primary hippocampal neurons	Co-incubation for 16 h	reduced TRPC6-dependent neuronal SOCE in postsynaptic spines increased mushroom spines percentages	[24]

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
