# Peer review of "Potential Drug Candidates to Treat TRPC6 Channel Deficiencies in the Pathophysiology of Alzheimer’s Disease and Brain Ischemia"

_cells, 2020, doi:10.3390/cells9112351_

Round 1

Reviewer 1 Report

The review manuscript “Potential drug candidates to treat TRPC6 channel deficiencies in the pathophysiology of Alzheimer’s disease and brain ischemia” by Veronika Prikhodko et al. summarize current known drug candidates, working as TRPC6 ion channel modulators to treat neurodegenerative disorders Alzheimer’s disease and cerebral ischemia. Although a lot of information has been listed in the manuscript, it was not well organized and the writing is poor.

General comments:

  1. To increase the readability for the general audience, abbreviation’s full name should be give before their use.
  2. Stay focused. I found it hard to follow when some of the sections drifted off the main topic- TRPC6. Authors spent quite a lot effort talking about the effects of drug candidates on AD or brain ischemia without mentioning TRPC6 at all.
  3. Authors should cite proper reference(s) for each major statement or clearly state that based on their own experiment or judgment.
  4. The terms should be consistent throughout the manuscript.

Specific suggestions/comments/questions:

Page 1, line 18: channels > channel

Page 1, line 19: channels > channel

Page 1, line 20: are > is

Page 1, line 27: The present review > this review

Page 1, line 27: provides information > summarizes the current available information

Page 1, line 28: TRPC6 channels > TRPC6 channel

Page 1, line 35: delete “to be”

Page1, line 35-36: in the following up decades: not clear what the authors want to say

Page1, line 40: toxic agents > toxic changes of proteins

Page2, line 50: affected brain volumn > brain volume affected

Page2, line 52-57: any references?

Page2, line 57: NMDAR > NMDA receptors

Page2, line 58: stage AD patients > stage of AD patients

Page2, line unsuccessful results in clinical practice and trials > unsuccessful clinical trials

Page2, line 60: NMDAR > NMDA receptors

Page2, line 70-73: based on a hypothesis to further speculate the possible mechanism of delaying the onset of the disease seems not solid and could be misleading

Page2, line 76-77: treatment options > treatments

Page2, line 78: The present review > this review

Page2, line86-93: any references?

Page2, line87: that have been described and > that can be

Page2, line 93: pseudogene in a large number of > pseudogene in

Page3, line 96: expression of TRPC6 is highly enriched > TRPC6 is highly expressed

Page3, lines 100-102: any references?

Page3, line 102: especially or > especially for

Page3, line 108: ER stores > Ca2+ stored in ER

Page3, line 116: TRPC6s play > TRPC6 plays

Page3, line 118: what does it mean “to the control level”?

Page3, line 131: nSOCE: either use neuronal SOCE or define nSOCE before using it. Same for the rest of nSOCEs in the rest of the text.

Page3, line 138: TRPC6s also act > TRPC6 also acts

Page4, line 143: There is evidence obtained by several research labs that > There is evidence that

Page4, line 143-145: any references?

Page4, lines 145-146, and in the rest of the text: fAD ?

Page4, line 147: suggest > suggested

Page4, line 147: that the mechanism of TRPC6 > that TRPC6

Page4, line 148: rather it involves > rather involves

Page4, line 149: PS2 rather with > PS2 with

Page4, lines 155-158: these two seem contradict each other, please explain.

Page4, line 165: demonstrate > demonstrated

Page4, line 166: indicate > indicated

Page4, line 167: associate > associated

Page4, line 168: bear > bore

Page4, lines 179-180: any references?

Page4, line 180: TRPC6s > TRPC6

Page4, line 186: Supporting this notion > In supporting this notion

Page5, line 230: These channels > This channel

Page6, line 243: TRPC6s > TRPC6

Page6, line 265: 20_HETE > 20-hydroxy-5Z,8Z,11Z,14Z-eicosatetraenoic acid

Page6, line 268: if 20-HETE did not affect intracellular Ca2+ concentration, how the threefold increase in TRPC6 activity was measured?

Page6, line 284-285: AT-NPD1 concentrations were > AT-NPD1 concentration was

Page7, line 288: 7 d following…… What does the d means?

Page7, line 293: Role of NPD1….. > The role of NPD1……

Page7, line 302: TRPC6s > TRPC6

Page7, line 315: expression levels > expression level

Page7, line 316: TRPC6 channels were > TRPC6 channel was

Page7, line 322: had no effect no > had no effect on

Page7, line 327: TRPC6 channels, and do not exert similar effect on its closest relatives, the TRPC3 channels > TRPC6 channel, and do not exert similar effect on its closest relative, the TRPC3 channel

Page7, line 328: activates TRPC6s and increases their expression > activates TRPC6 and increases its expression

Page7, line 331: have been > has been

Page7, line 332: In rats injected with amyloid fibrils ? What was injected to the rats? Amyloid fibrils or hyperforin? I guess is hyperforin, but the sentence is confusing.

Page7, line 333: specie > species

Page 8, line 368: By Shin et al. resveratrol monotherapy was reported > Resveratrol monotherapy was reported by Shin et al.

Page8, line 369: up until > up to

Page8, line 369: of ischemia > of ischemia.

Page8, line 370: rather than > than that

Page8, line 373: cerebrovascular events > cerebrovascular conditions

Page8, line 376: Another trial found resveratrol to significantly increase diastolic blood pressure and reduce > Another trial found that resveratrol significantly increased diastolic blood pressure and reduced

Page9, line 391-392: TRPC6s > TRPC6

Page9, line 411, TRPC6 channels > TRPC6 channel

Page9, lines 413-414: in 6 month old hippocampal slices taken from 5xFAD mice > in hippocampal slices taken from 6 month old 5xFAD mice

Page9, lines 415-420: what is the relationship between APP695 and AZP2006? The sentence is so confusing.

Page9, line 421: FDES > Bis-{2-[(2E)-4-hydroxy-4-oxobut-2-enoyloxy]-N,N-diethylethanaminium} butandioate

Page10, line 437: 1 h > 1 hour

Page10, line 439: NSN21778 >  N-[4-[2-[(6-aminoquinazolin-4-yl)amino]ethyl]phenyl]acetamide

Page10, line 444: it was found to > NSN21778 was found to

Page10, line 447: TRPC6 channels > TRPC6 channel

Page10, lines 449-450: be considered of interest as > be considered as

Page10, line 454-455: piperidine-1-carboxylate) > piperidine-1-carboxylate

Page10, line 455: was the  most active > was most active

Page10, lines 455-457: Please explain why Compound 4n (ethyl 453 4-(7-hydroxy-2-methyl-3-(4-(trifluoromethyl)phenyl)-pyrazolo[1,5-a]pyrimidin-5-yl)piperidine-1-ca454 rboxylate was most active towards TRPC6  with lower EC50 values.

Page10, line 462: and independently > and independent

Page10, line 465: exposed to > with

Page11, lines 488-496: What is the relationship between TRPC6 inhibition and 20-HETE inhibition?

Page11, line 511: All of the compounds > These compounds

Page11, line 515: 24 h-long > 24 hour-long

Page12, line 561: chemical structures > chemical substances

Reviewer 2 Report

The manuscript by Prikhodko discusses potential drug candidates for targeting TRPC6 receptor protein in order to treat either Alzheimer’s Disease (AD) or ischemia stroke. TRPC6 is a non-selective cation channel in the plasma membrane that is permeable to Ca+2. Because Ca+2-mediated excitotoxicity is implicated in the neuronal cell death mechanisms for both AD and ischemic stroke, the authors argue that TRPC6 is a good candidate for screening/developing small molecule drug candidates that can improve neuroprotection in AD and ischemic stroke. The authors explain the role of Ca+2 in neuronal excitotoxicity and note studies several studies implicating TRPC6 in neuroprotective pathways. In this discussion, however, the authors also note that there is conflicting evidence as to whether TRPC6 function is either up- or down-regulated in AD or ischemic stroke. The review concludes with a survey of previously identified compounds that either activate or antagonize TRP6C function by either direct or indirect mechanisms, and whether these compounds have been tested in either AD or stroke model systems.

In general, the manuscript is well organized and written. The major issue with manuscript is the focus on TRPC6 itself. The authors present the role of TRPC6 as controversial, which raises the question of why spend the time and expense of exploring small molecules/pharmacological agents to modify TRPC6 function if it is unclear whether one should be activating or repressing TRPC6 function in order to enhance neuroprotection? From the literature presented, however, there seems to be a preponderance of evidence indicating that TRPC6 function is down-regulated in AD and ischemic stroke, and there is a clear need for compound that can increase/enhance TRPC6 function in order to improve neuroprotection. The authors should revise their presentation of the controversy in the literature and more critically assess the credibility/limitations of studies that suggest that TRPC6 function is down-regulated in AD and ischemic stroke.

A second major issue not addressed by the authors is the role of potential off-target interactions by the compounds they survey. Some these compounds can interact with other protein targets and the authors should discuss whether these off-target interactions can interfere with an ability to assess the impact on TRPC6 function.

A minor issue is that some of the English needs to be revised for more appropriate word choices. The first sentence, for example, states, “…cerebrovascular diseases are called by WHO to be the main cause of disability in the following up decades.” This would better stated as, “…cerebrovascular diseases are considered by WHO to be the main cause of disability in the coming decades.”

Round 2

Reviewer 2 Report

The authors have satisfactorily addressed my concerns regarding initial manuscript submission and it is suitable for publication.